# Antioxidant Efficacy and “In Vivo” Safety of a Bentonite/Vitamin C Hybrid

**DOI:** 10.3390/pharmaceutics15041171

**Published:** 2023-04-07

**Authors:** Dayaris Hernández, Anaela Montalvo, Irela Pérez, Clarence Charnay, Rita Sánchez-Espejo, Pilar Cerezo, César Viseras, Serena Riela, Giuseppe Cinà, Aramis Rivera

**Affiliations:** 1Zeolites Engineering Laboratory, Institute of Materials Science and Technology (IMRE), University of Havana, Havana 10400, Cuba; 2Department of Drugs Technology and Control, Institute of Pharmacy and Food (IFAL), University of Havana, Havana 13600, Cuba; 3Institut Charles Gerhardt de Montpellier, CNRS UMR 5253, Université de Montpellier, 34095 Montpellier, France; 4Department of Pharmacy and Pharmaceutical Technology, Faculty of Pharmacy, University of Granada, Campus of Cartuja, 18071 Granada, Spain; 5Andalusian Institute of Earth Sciences, CSIC-University of Granada, Avda. de Las Palmeras 4, 18100 Armilla, Spain; 6Department of Biological, Chemical and Pharmaceutical Sciences and Technologies, University of Palermo, Viale delle Scienze, Ed. 17, 90128 Palermo, Italy

**Keywords:** bentonite, L-ascorbic acid, antioxidant capacity, photoprotection

## Abstract

L-ascorbic acid (LAA), commonly known as vitamin C, is an excellent and recognized antioxidant molecule used in pharmaceutical and cosmetic formulations. Several strategies have been developed in order to preserve its chemical stability, connected with its antioxidant power, but there is little research regarding the employment of natural clays as LAA host. A safe bentonite (Bent)—which was verified by in vivo ophthalmic irritability and acute dermal toxicity assays—was used as carrier of LAA. The supramolecular complex between LAA and clay may constitute an excellent alternative, since the molecule integrity does not seem to be affected, at least from the point of view of its antioxidant capacity. The Bent/LAA hybrid was prepared and characterized through ultraviolet (UV) spectroscopy, X-ray diffraction (XRD), infrared (IR) spectroscopy, thermogravimetric analysis (TG/DTG) and zeta potential measurements. Photostability and antioxidant capacity tests were also performed. The LAA incorporation into Bent clay was demonstrated, as well as the drug stability due to the Bent photoprotective effect onto the LAA molecule. Moreover, the antioxidant capacity of the drug in the Bent/LAA composite was confirmed.

## 1. Introduction

L-ascorbic acid (LAA) is an active ingredient used in dermocosmetic formulations as being antioxidant, anti-aging, wound-healing, photoprotecting, and anti-pigmentary [1]. This essential nutrient known as vitamin C has vital functions in humans, who lack the ability to synthesize their own LAA due to the deficiency of an enzyme. Its presence in the individuals helps the growth and maintenance of healthy bones, teeth, gums, ligaments, and blood vessels, and it is involved in important metabolic functions [2,3]. LAA, probably, is the most plentiful and the most important water-soluble radical-trapping antioxidant in biological systems [1,4]. Water solubility of this vitamin enables the regeneration of other antioxidants such as alpha-tocopherol (vitamin E). As an ultraviolet (UV) photoprotection agent, it also has a synergistic effect when is used in conjunction with vitamin E [5]. LAA ability to reduce skin wrinkles is associated with an increase of fibroblasts promoting collagen synthesis [6]. Its anti-pigmentary effect is given by the interaction with copper ions at the active site of the tyrosinase enzyme, inhibiting its role in converting tyrosine into melanin [1].

Three basic factors affect the stability of LAA in aqueous solution: high pH, the presence of oxygen and the metal ions [7]. Under aerobic conditions, LAA’s degradation process begins with two reversible steps that lead to the formation of dehydro-L-ascorbic acid (DHLAA), which irreversibly degrades to 2,3-diketo-L-gulonic acid (2,3-DKLGA) [8]. The final degradation products have been identified as 2-furoic acid and 3-hydroxy-2-pyrone [9,10]. Despite its importance in biomedical applications, the inclusion of LAA in different preparations has been discarded due to its major instability [4]. In order to solve it, i.e., to maintain the lifetime of vitamin C in the formulations, several proposals have been developed. The simplest strategies to mitigate LAA degradation involve controlling the presence of oxygen during formulation and storage, working with low pH, reducing water content and using preservatives [7]. Other chemical solutions are related to obtaining derivatives, but no compound obtained becomes the ideal substitute. Given these scenarios, the incorporation of LAA in a carrier system should be a suitable solution to improve its stability problems. Several systems have been proposed: nanoencapsulation of Vitamin C in liposomes, mesoporous silica nanoparticles, polymeric nanoparticles, microemulsions, micelles and clay minerals [4,7].

Based on the advantageous properties of the clays and clay minerals―ion exchange, adsorption, swelling, and rheological behavior―and their use as support [11,12,13,14,15], the pharmaceutical and cosmetic industries have an increasing interest in incorporating these materials to their products. In this sense, a special emphasis has been made in natural resources by their abundance, low cost, and environmental friendliness [16,17,18,19]. Thus, it is worth underlining the employment of clays in skin care formulations [20,21,22]. The protective effect of clays when used as support in order to stabilize drugs in formulations has also been reported [22,23]. In particular, bentonite, a material mainly composed of the mineral montmorillonite, arises as an interesting carrier for LAA. LAA incorporation in montmorillonite was reported, although no evidence of drug photoprotection and antioxidant capacity was established for the composite material [8,24,25] (see Appendix A).

The starting point of the present work is to take advantage of the use of bentonite (Bent) as a drug host to preserve the LAA antioxidant capacity, as well as the potential of this clay in topical formulations. Toxicological and mineralogical studies on Bent were performed. A Bent/LAA hybrid was prepared, characterized, and its photostability and antioxidant capacities were determined. 

## 2. Materials and Methods

### 2.1. Materials

Cuban natural bentonite (Bent) clay was used as raw material, which was supplied by the Research Center for Mining–Metallurgical Industry (CIPIMM). LAA (C_6_H_8_O_6_), pharmaceutical grade according to the United States Pharmacopoeia [26], was the drug evaluated. It was used as received from the Cuban pharmaceutical industry. All other chemicals used in the study were analytical grade.

### 2.2. Methods

#### 2.2.1. Preparation of Raw Clay

Firstly, the raw bentonite was submitted to a further purification process, where 50 g of clay was suspended in 1 L of distilled water under magnetic stirring at 800 rpm for 1 h. After that, the dispersion was left to stand for 0.5 h and the first transparent supernatant, which represents more than half of the initial water volume, was eliminated by decantation. The second contains the clay/water dispersion. Purification was repeated until no iron impurities on the surface of the stirrer’s magnetic bullet were observed. Finally, the dispersion was centrifuged and the solid resulting was oven-dried at 100 °C for 24 h and stored at room temperature.

#### 2.2.2. Toxicological Studies

To evaluate the safety of the Bent when it is topically administrated to experimentation animals, ophthalmic irritability and acute dermal toxicity studies were performed according to the procedure described by the Organization for Economic Co-operation and Development (OECD TG 405 and 402) [27,28] and approved by the Institutional Ethics Committee of the Institute of Pharmacy and Food (IFAL), University of Havana, Cuba (Protocol code IFAL-CEI 122/124-2021, 12 May 2021).

The animals with certified health were supplied by the Center of Laboratory Animals (CENPALAB, La Havana, Cuba). The quarantine conditions were: 22 ± 3 °C of temperature, relative humidity in the range 30–70 ± 5% and light/darkness cycle of 12/12 h. Water and food were supplied ad libitum. Male New Zealand albino rabbits with a corporal weight between 1.8 and 2.0 kg and Wistar albino rats with a corporal weight between 200 and 270 g were employed for ophthalmic irritability and acute dermal toxicity tests, respectively. The animal study protocol was approved and established by the Animals Care Committee of the Center of Study for the Research and Biological Evaluation (CEIEB) of the Institute of Pharmacy and Food, University of Havana, Cuba. The tests were performed following ethical guidelines for animals regarding the use of the smallest possible number of individuals, and causing minimum suffering, in the spirit of the principle of reduction and refinement [29]. In order to facilitate the achievement of these goals, only the main component (Bent) of the supramolecular complex (Bent/LAA) was evaluated.

In the ophthalmic irritability study, an exhaustive evaluation of the ocular structure of one group of rabbits (n = 3) was carried out before starting the assay. A mass of 0.1 g of the clay was applied in the conjunctival cul-de-sac of the right eye, while the left eye served as the negative control. One hour later, the conjunctiva, iris, and cornea ocular structures were examined. This took place individually for each animal at 1, 24, 48, and 72 h after the administration according to the Draize scale [30] for this assay.

For acute dermal toxicity, two groups of rats (n = 5) of both sexes were studied. In order to use the correct dosage, the animals were weighed before starting the study. Twenty-four hours before the start of the test, 10% of the back corporal surface of the animals was shaved. After applying a dose of 2000 mg/Kg of Bent, the area was covered with sterile gauze and the whole region was protected with a hypoallergenic rubber band. The clay was left in contact with the skin for 24 h, then the cover was removed using an isotonic saline solution with the help of a gas. During the first day the animals were observed individually and frequently, and then observations were made daily for the remaining 13 days for presence of any edema, erythema, or any type of dermal change. Rats were weighed during days 1, 7, and 14.

At the end of the assays, the animals were euthanized with an over-dosage of sodium thiopental. If any affectation was found during the observations performed on the organs (lungs, heart, spleen, kidneys, and stomach), then samples were submitted to a histopathology study.

#### 2.2.3. Preparation of Clay/Drug Composite

Ten mL of LAA aqueous solution at 9 mg/mL was put in contact with 100 mg of Bent in powder and continuous stirring during 24 h at room temperature. After that, the dispersion was centrifuged for 10 min at 1000 rpm. In order to protect the Bent/LAA suspensions from the effect of light, amber containers with aluminum paper as extra protection were used. At the end of the process and after centrifugation, a total transparency of the supernatant was also observed. It confirms the non-degradation of the drug since no color changes were evidenced. Moreover, with the aim to guarantee the thermal stability of the drug in the Bent/LAA nanocomposites, they were dried at 80 °C for 1 h on glass plates. The LAA concentration in the supernatant after interaction was analyzed and quantified by ultraviolet (UV) spectroscopy (Rayleigh UV-2601 spectrophotometer, Beijing, China) in the wavelength interval 200–400 nm. It was performed at λ_max_ = 243 nm using the Lamber–Beer law and a calibration curve, according to standard procedures reported in the pharmacopeia [26]. The amount of LAA adsorbed on the Bent (i.e., adsorbent loading) was calculated as follows:(1)$$\;q_{e}=\frac{(C_{o\;}-C_{f})\times V}{m}$$
where *q_e_* (mg/g) is the mass of adsorbed LAA per unit mass of the adsorbent, *C_o_* is the initial drug concentration (mg/mL), *C_f_* is the concentration of the LAA solution at the equilibrium (mg/mL), *V* is the volume of the solution (mL), and m is the mass of Bent (g) used in the experiments.

#### 2.2.4. Physical–Chemical Characterization

The purified clay sample (Bent) was analyzed using a wavelength-dispersive X-ray fluorescence (WDXRF) spectrometer (model Axios Max from PANalytical, Almelo, The Netherlands) on fused glass beads. To obtain high resolution spectra, eight dispersive LiF200, LiF220, PE002, Ge111, PX1, PX4a, PX5, and PX7 crystals were employed. The oxide-form composition was calculated using the fundamental parameters method.

For the photostability tests, 60 mg of Bent/LAA powder were put in a glass sample holder. The sample was introduced into a large UV darkroom Vilber CN-15.LC. It was irradiated for a maximum time of 8 h with four 15-watt UV tubes, two at 254 nm and two at 365 nm, simultaneously. Samples were collected at intervals of 2 h and characterized by X-ray diffraction (XRD).

X-ray diffractograms of the samples in powder form were collected in a X’pert diffractometer (Philips Analytical, Almelo, The Netherlands)with Cu Kα radiation (λ = 1.54 Å) at room temperature, operating at a voltage of 45 kV and working current 25 mA. The studies were done at a scan rate of 1°/min for a 2θ range from 3 to 70°.

Thermogravimetric (TG/DTG) analysis of the raw material and the clay/drug nanocomposite was carried out with the aid of a thermal analyzer (Perkin Elmer STA 6000, Waltham, MA, USA), using nitrogen atmosphere at a heating rate of 10 °C/min from 30 to 800 °C. The sensitivity of the thermobalance was ±1 μg. A solid sample of about 15 mg was used in each test.

Fourier transform infrared spectroscopy (FT-IR) spectra were recorded using a spectrometer (Perkin Elmer Two, Waltham, Massachusetts, United States) equipped with a universal attenuated total reflection (UATR) accessory. The measurements were carried out in the spectral range from 4000 to 400 cm^−1^. The force gauge applied for each solid sample was 85% to ensure good sample–crystal contact.

Zeta potential (ZP) measurements were performed using a Zetasizer Nano ZS (Malvern Instruments, Malvern, UK). The samples were dispersed in deionized water (with a conductivity around 0.056 µS/cm) and the pH was adjusted by dropwise addition of NaOH or HCl solutions at 2 M. The suspensions were prepared adding 1 mg of powder in 2 mL of water and put in an ultrasonic bath for 15 min. The measurements were carried out at room temperature, and every sample was measured three times and the average values were reported.

#### 2.2.5. Antioxidant Capacity

LAA molecule acidity results from the presence of an enediol group in its structure (C2 and C3, see Figure 1), which is responsible for the LAA antioxidant activity [31]. The radical-scavenging activity of the Bent/LAA composite was determined by a spectrophotometric procedure using the free radical 2,2-diphenyl-1-pricrylhydrazyl (DPPH^●^) method at 513 nm [32]. The positive reaction between an antioxidant compound and the DPPH^●^ results in a decoloration of the DPPH^●^ solution from violet to yellow [33]. The Bent/LAA nanocomposite was dispersed in methanol at different concentrations. One mL of DPPH^●^ solution (1.5 × 10^−4^ M) in methanol was then added to the tested sample. The mixture was then vigorously stirred and allowed to stand at room temperature in the dark for 40 min. The inhibition of free radical DPPH^●^ in percent (I%) was calculated as:(2)$$I\%=\left[\frac{\left(Abs_{0\;}-Abs_{s}\right)}{Abs_{0}}\right]\times 100$$
where *Abs_0_* is the absorbance of the DPPH^●^ solution without sample and *Abs_s_* is the absorbance of the test sample. 

The antioxidant activity (IC_50_) is defined as the sample concentration that reduced the initial DPPH^●^ absorbance to 50% [32]. Thus, from plotting the percentage of inhibition against concentration, the curves are obtained. From the equation of this curve, the value of IC_50_ is determined. A smaller IC_50_ value corresponds to a higher antioxidant activity. All test analyses were realized in triplicate. For comparison, the antioxidant activity of the pristine compounds, Bent and LAA, were also evaluated. 

## 3. Results and Discussion

### 3.1. In Vivo Toxicity Studies

During the first day of the ophthalmic irritability test in rabbits, secretions in the conjunctiva were observed, as well as edematous and erythematous reactions, which disappeared in the subsequent days. No lesions in the iris and cornea were detected. No other clinical signs of toxicity were perceived in the animals when the clay was administered to the ocular structure. The results indicated an irritability index of 2.0 points based on scale scores for grading the severity of ocular injuries [30]. It allows classification of the Bent as no irritant (see Appendix A).

The results of the acute dermal toxicity assay, for both groups of rats, demonstrated no skin lesions or other clinical signs indicative of a toxic effect (see Appendix A). Bent did not affect the weight gain of the animals under study, which suggests the absence of systematic toxic effects. At the time of the necropsies, the samples of the selected organs did not show any alteration from the macroscopic point of view. Thus, the histopathology study did not take place. Bent did not produce dermal acute toxicity in the experimentation animals following the assay described by the OECD TG 402 [28] with a dose of 2000 mg per kg of corporal weight, classifying it as “without classification” according to the European Union. 

### 3.2. Characterization of Raw and Composite Materials

The chemical composition determined by XRF of the major oxides for the purified Bent sample is shown in Table 1. The mineralogical formula of the clay was estimated based on these results and from qualitative phases composition revealed by X-ray diffraction (see below Figure 2a). It was calculated using the stoichiometry of the representative composition. The result expressed in the half unit cell was: $$\left({\mathrm{Na}}_{0.44}{\mathrm{Ca}}_{0.07}K_{0.02}\right)\left({\mathrm{Al}}_{1.12}{\mathrm{Fe}}_{0.61}^{3+}{\mathrm{Mg}}_{0.28}{\mathrm{Ti}}_{0.05}\right)\left({\mathrm{Si}}_{3.45}{\mathrm{Al}}_{0.55}O_{10}\right){\left(\mathrm{OH}\right)}_{2}$$

The composite was prepared following the methodology described in Section 2.2.3: the drug load determined by UV spectroscopy was around 450 mg ± 50 of LAA per gram of Bent. The pH value of the clay/drug suspension spontaneously reached was around 3.5. It is convenient, since the largest LAA molecules stability takes place at acid pH below its pKa_1_ value (4.2) [34], and where the LAA is mainly in a non-ionized form (H_2_LAA). Thus, it is an important criterion for a future escalation process.

X-ray diffraction patterns for Bent and LAA pristine compounds, and the Bent/LAA composite, are shown in Figure 2a. Montmorillonite (Mt) is the main mineralogical phase, which was identified following the most intense reflections labeled as Mt [35]. Quartz (Q) and calcite (Cal) were the other phases identified [35]. For LAA, the XRD pattern is typical of a crystalline material, which is coherent with previous reports [36,37]. Bent exhibits a reflection at 2θ = 7.0° corresponding to 001 basal reflection of the Mt phase. For this angle, according to Bragg’s Law, the inter-planar distance (*d*-value) was 1.26 nm. It agrees with that reported in the literature for a natural Mt containing simultaneously sodium and calcium cations with one layer of water intercalated in the interlayer space [38,39]. For the Bent/LAA composite, the 001 reflection around 2θ = 5.6° corresponding to a *d*-value of 1.57 nm was identified. When both patterns (Bent and Bent/LAA) are compared, a shift to lower angles is detected in the raw material after the interaction with the drug. It is known that the thickness of a Mt single layer is around 0.94 nm [40]. For the Bent/LAA hybrid, a basal spacing increase of around 0.31 nm is observed (Δ*d* = 0.63 nm, aperture of the interlayer space) due to the LAA presence. 

If we consider the dimensions of the unit cell of Mt (*a* = 0.52, *b* = 0.90, *c* = 1.00 nm [41]), those of the LAA molecule (0.74 in length, 0.50 wide, and 0.34 nm in thickness following the more stable conformers [42]), the existence of free rotation carbons in the drug structure, as well as the variation of the *d*-value in the XRD pattern of the composite, we conclude that one LAA molecule per unit cell of Mt can easily accommodate in the interlayer space. On the other hand, if we consider that 1 g of Bent contains 7.69 × 10^20^ unit cells and that the LAA load measured by UV spectroscopy is about 450 mg per gram of Bent, in one Bent unit cell, there are 5.85 × 10^−19^ mg of LAA. Then, if one LAA particle weighs 2.93 × 10^−19^ mg, we would have around 2 LAA molecules per unit cell. So, we hypothesize that at least one of these two LAA molecules is located in the interlayer space, and the other is adsorbed on the clay surface and into the material mesoporosity/macroporosity. This scenario semi-quantitatively explains the relatively large load of LAA on Bent.

Now, let us examine some of these ideas in some more detail. A representation of the possible accommodation of LAA molecules on the clay is shown in the right panel of Figure 3. We are assuming that the LAA molecules intercalated in the interlayer space are oriented in such a way that its smallest dimension is perpendicular to the clay layers. Other molecules, however, can block the interlayer space by the preferential orientation of the enediol group and the other hydroxyl groups of the drug through hydrogen bonds and Van der Waals forces, respectively. Moreover, it should be taken into account that during the interaction process clay/drug, the water molecules can be also located between lamellae, contributing to increasing the interlayer space by the effect of two water layers intercalated [38,39]. In summary, the results point to adsorbent/adsorbate interactions that could have both chemical and physical interactions.

The diffraction patterns for Bent/LAA composite at 6 h and 8 h of irradiation are shown in Figure 2b. No variations in the XRD patterns for the irradiated samples during 2 and 4 h were observed. The comparison of the nonirradiated Bent/LAA composite and those irradiated for 6 and 8 h evidenced a slight decrease in the basal spacing (*d*001) with the irradiation time. At 6 h and 8 h, the *d*001 values were around 1.51 and 1.47 nm, respectively, which could indicate that a slight degradation of the intercalated LAA occurred. It should be noted that this is not representative of the decay rate of the LAA, especially when dealing with intercalated species. In this sense, the effect of Bent on the LAA decay rate has been studied for stability enhancement of the drug [8]. In that case, the LAA concentration in solution for several Bent amounts as a function of time was examined. The results indicated that when the Bent is in the dissolution medium, the LAA decomposition in dissolution noticeably decreases. All in all, our results suggest the Bent photoprotective effect onto LAA structure when the composite was exposed to UV radiation. Similar effects have been reported for tetracycline/clay minerals systems [43], where the use of clays as a promising materials for the drug photoprotection was demonstrated. In addition, there are studies that have revealed the pronounced photoprotection of glycine and adenine molecules by the nontronite clay present on the sample [44].

In addition, the small peak corresponding to Cal (104 reflection plane) in the XRD pattern of the Bent/LAA hybrids (nonirradiated and irradiated samples) diminishes due to the effect of the acid medium.

FTIR spectra for LAA, Bent, and Bent/LAA composite using the attenuated total reflectance (ATR) mode are displayed in Figure 4, indicating the wavenumbers of their characteristic bands. In the spectrum of Bent appear the main bands associated with the normal O-H and Si-O vibration modes, distinctive in this material. Typical valence vibrations of the structural hydroxyl groups for dioctahedral smectite and adsorbed and intercalated water molecules in the clay ($\nu_{O-H}$ about 3622 and 3408 cm^−1^, respectively) were identified. The band at 1637 cm^−1^ can be attributed to the bending vibration in the plane of water ($\delta_{O-H}$). The intense band about 985 cm^−1^ corresponds to the Si-O stretching vibrations of the tetrahedral sheet ($\nu_{Si-O}$) (see Figure 4) [17,45,46,47]. Lastly, the signal around 1456 cm^−1^ is due to stretching vibration of carbonate anion ($\nu_{CO_{3}^{2-}}$) as a spurious phase present in the raw sample [17,46].

The LAA spectrum (Figure 4) shows four bands at 3521, 3402, 3303, and 3201 cm^−1^ associated with stretching vibration of the hydroxyl groups ($\nu_{O-H}$). At 3001 and 2914 cm^−1^ appears the Csp^3^-H stretching associated with CH and CH_2_ groups $\left(\nu_{Csp^{3}-H}\right)$. The signal at 1753 cm^−1^ was assigned to the valence vibration C=O of the carbonyl group ($\nu_{C=O}$) of the five membered alpha lactone ring system. The intense band at 1650 cm^−1^ arises from C=C stretching vibrations ($\nu_{C=C}$ ring stretching) [48,49]. Others vibrational bands are observed in the region 1500–1000 cm^−1^, which are connected with the C-H deformation modes (CH_2_ scissoring and wagging, and bending of the CH group), C-O-C stretching and C-O-H bending [48,50].

In the Bent/LAA composite spectrum (Figure 4), a wide band was observed in the region from 3750 to 2400 cm^−1^ with a center at 3622 and 3371 cm^−1^. It is associated with the O-H valence vibration ($\nu_{O-H}$), where drug signals appear overlapped with that clay. It this sense, in the Bent/LAA spectrum, the four bands of the LAA mentioned above provoked deformations in the clay band. It provoked a shift to lower frequencies in the Bent/LAA in respect to Bent (from 3408 to 3371 cm^−1^). The peak at 1631 cm^−1^ in the composite material is due to the overlapping of C=C stretching vibrations ($\nu_{C=C}$) and to a lesser extent, to the water bending vibration ($\delta_{O-H}$). A slight change to lower frequencies is also observed in this band when comparing the composite spectrum with those of the raw materials. Thus, the changes observed to the lower wavenumber suggest weak clay/drug interactions. The band at 985 cm^−1^ corresponds to valence vibration Si-O of the tetrahedral sheet ($\nu_{Si-O}$). The signal associated with the carbonate is missing in the composite spectrum, due to the clay/drug interaction taking place at pH around 3.5. It results in the dissolution of the calcite spurious phase, which is consistent with XRD analysis.

TG/DTG curves for Bent, LAA, and Bent/LAA composite are shown in Figure 5. For Bent, three weight loss peaks were observed (see DTG curve): the first main mass loss around 85 °C corresponds to the dehydration of interparticle water, adsorbed and interlayer water [51,52,53,54]. The second weight loss event from 380 to 580 °C with a maximum at 484 °C, as can be seen in the DTG curve, is due to the removing of the hydroxyl groups from the laminar structure in water form (structural dehydroxylation) [55,56,57]. The last smaller weight loss about 671 °C is associated with another stage of dehydroxylation [58] that can lead to the total structure collapse [55,57]. These results are in good agreement with the previous studies reported in the literature [17,59,60,61].

From the TG/DTG curves (Figure 5) of the ascorbic acid, it can be seen that the dry solid is thermally stable and its decomposition begins from 180 °C. The TG curve also indicates that the decomposition process can be divided into three stages [62,63]. The first mass loss around 228 °C corresponds to dehydration and decarboxylation reactions, which takes place from 180 to 258 °C. The second one from 258 to 410 °C (peak at 310 °C in the DTG curve) is associated with decarboxylation and decarbonylation of the organic molecule. Finally, at the third stage at 410–610 °C, with a maximum at 518 °C, as can be seen in the DTG curve, only a carbonization process of the drug occurs [62].

In the DTG curve (Figure 5) of the composite material Bent/LAA, three mass losses were identified about 90, 243, and 478 °C. The first one corresponds to water loss of the material. The next peak between 196 and 350 °C could be assigned to the second step of the decomposition of intercalated and/or adsorbed LAA which confirms its presence in the composite. The last weight loss from 350 to 800 °C could be associated to the last step of LAA decomposition and the final dihydroxylation of the clay. The comparison of the DTG curves of the LAA and the Bent–LAA composite evidenced that the LAA decomposition occurs at lower temperatures after the interaction with the clay. It suggests a decrease in the thermal stability of the drug in the composite, corroborating the existence of interactions between the LAA and the clay. These results are coherent with those by IR spectroscopy and X-ray diffraction.

Surface charge properties for Bent and Bent/LAA composite were analyzed from the ZP results (Figure 6). It is known that the zeta potential of Mt particles does not change significantly with the pH [64]. For both samples, the ZP values were negative and no isoelectric point (point of zero charge, pHpzc) was observed within the pH range studied. It indicates that the electrokinetic potential is dominated by the permanent layer charge in the montmorillonite [65]. From a qualitative point of view, the behavior for both samples (Bent and Bent/LAA) is very similar. However, the surface charge of the Bent/LAA material particles becomes less negative due to the presence of the LAA in the clay. From the quantitative point of view, the curve corresponding to the Bent/LAA is more electropositive than that of the starting material. It is more evident for pH 5 and 7, where the variations ZP values were −18.2 and −18.9 mV, respectively. Such changes in the surface charge may confirm the contribution of electrostatic interactions or Van der Waals forces in the composite material formation. For example, in non-symmetric molecules with polar groups, like the case of the LAA in non-ionized form (H_2_LAA), electronic densities are located on the more electronegative elements, generating dipoles which can interact with the Mt negative charge.

### 3.3. Antioxidant Activity

Figure 7 shows the evaluation of the DPPH^●^ activity (expressed in inhibition percent) for both LAA and Bent pristine compounds, and for Bent/LAA composite. The figure also sketches the reaction between LAA and the DPPH^●^. As expected, LAA showed a good antioxidant activity, which is preserved after its interaction with Bent, and it is proved by the significant decoloration of the DPPH^●^ solution. Pristine Bent displayed no changes in the DPPH^●^ radicals solution decoloration, i.e., it did not show antioxidant activity. These results are in agreement with the IC_50_ values calculated for each sample (Figure 7a). The higher antioxidant capacity to the LAA sample (IC_50_ = 11.24 μg/mL) followed by the Bent/LAA composite (IC_50_ = 2.5 mg/mL, which corresponds to an IC_50_ = 1.12 mg/mL in terms of LAA loaded into composite) was verified. The higher IC_50_ obtained in the case of Bent/LAA was explained taking into account that the LAA antioxidant activity is mainly due to the action of the –hydroxyls group (–OH) linked to C2 (Figure 7b). After interaction with Bent, this –OH could establish hydrogen bonds with clay and thus, it is not available anymore to exert the antioxidant activity. Clay–organic molecules interactions, via hydrogen bonds, are well known [66,67]. Therefore, we conclude that Bent performs some protective effect on LAA antioxidant activity.

## 4. Conclusions

In the present study, the safety of a Bentonite was demonstrated, so its use for pharmaceutical applications is recommended. The results allowed one to estimate the mineralogical formula of the clay and demonstrate the ability of Bent to incorporate LAA (up to 450 mg ± 50 per gram of clay). They also suggested LAA adsorption on the clay, as well as its intercalation and interlayer space blockage. The Bent photoprotection effect onto the LAA structure in the composite material was confirmed. The adsorbent/adsorbate interactions could have both physical and chemical natures, with a prevalence of electrostatic interactions. The IC_50_ values allowed verification of the antioxidant activity of the Bent/LAA hybrid, corroborating the protective action of the clay on the LAA molecule. The results strongly support the potential use of the Bent/LAA composite in skin care applications.

## Figures and Tables

**Figure 1 pharmaceutics-15-01171-f001:**
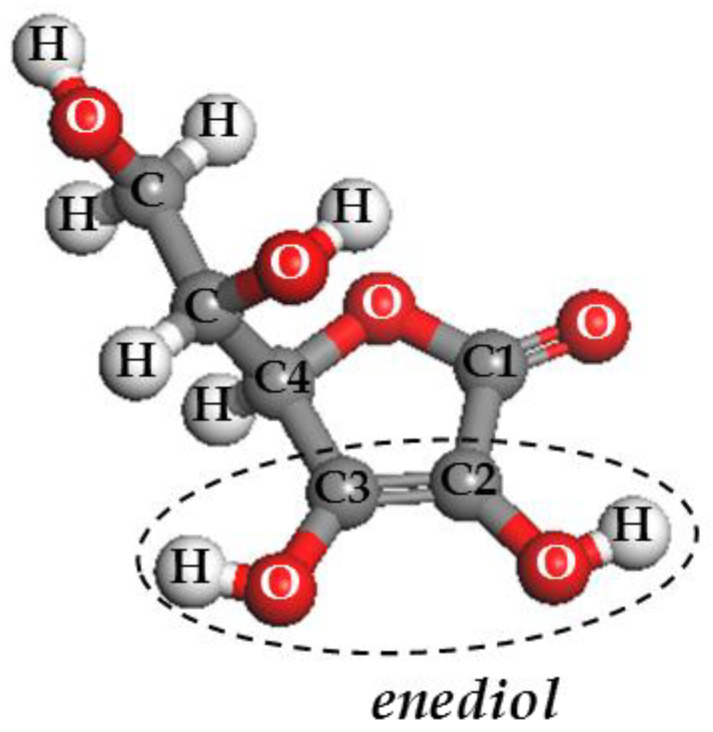
Molecular structure of the L- ascorbic acid. The enediol group (C2 and C3) with antioxidant capacity is encircled in a dashed line.

**Figure 2 pharmaceutics-15-01171-f002:**
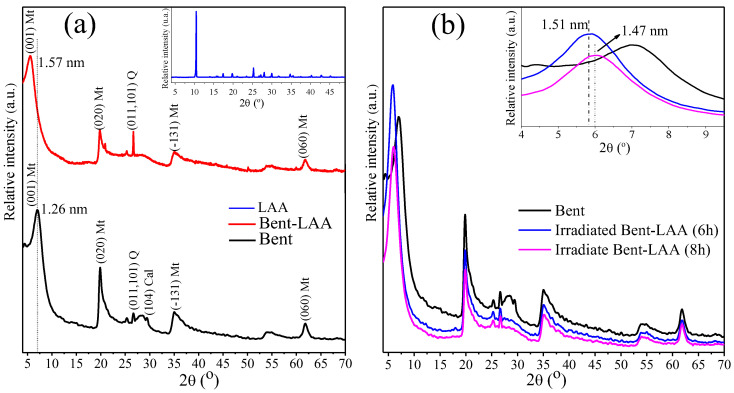
XRD diffractograms for (**a**) LAA, Bent, and Bent/LAA composite, indicating the *hkl* values of the characteristic planes of each phase; (**b**) Bent, Bent/LAA after 6 h and 8 h of irradiation. The inset shows a zoom that illustrates the small shift to lower *d*-values of the characteristic 001 basal reflection of the Mt phase after the samples were irradiated.

**Figure 3 pharmaceutics-15-01171-f003:**
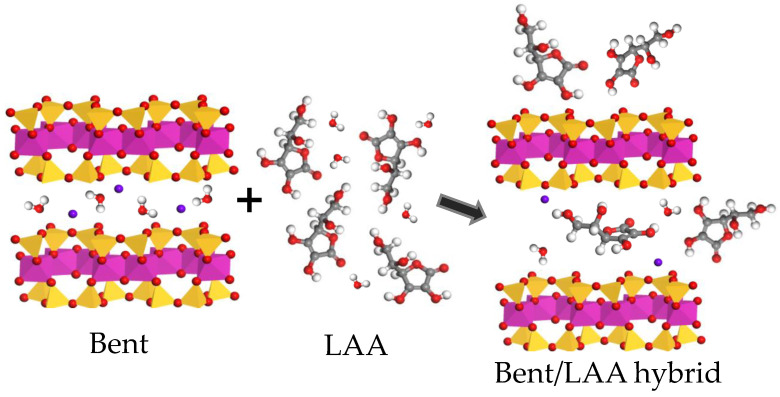
Schematic representation of the load of LAA on the Bent.

**Figure 4 pharmaceutics-15-01171-f004:**
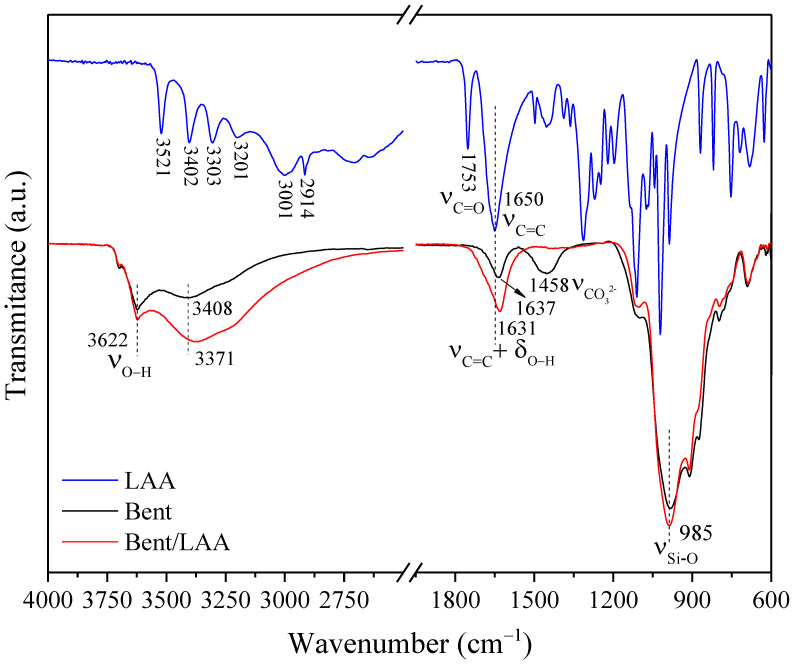
FTIR spectra of the samples LAA, Bent, and the Bent/LAA employing the ATR mode.

**Figure 5 pharmaceutics-15-01171-f005:**
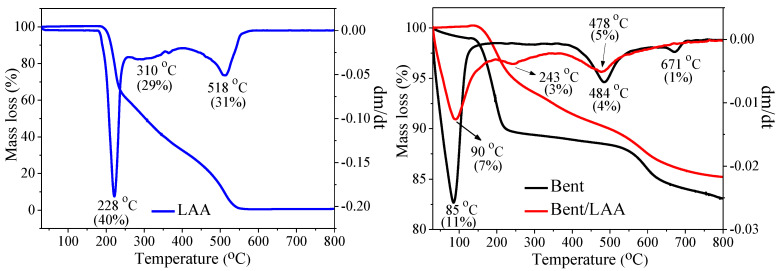
Thermograms (TG/DTG) for Bent, LAA, and Bent/LAA composite.

**Figure 6 pharmaceutics-15-01171-f006:**
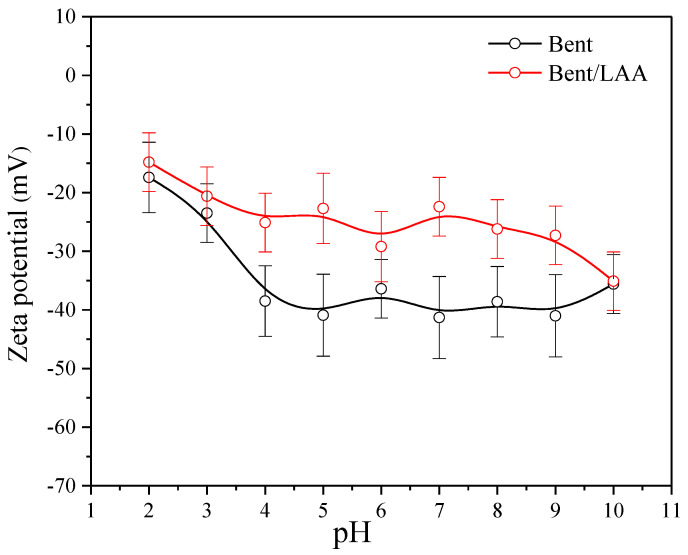
Zeta potential values of Bent and Bent/LAA composite as a function of pH.

**Figure 7 pharmaceutics-15-01171-f007:**
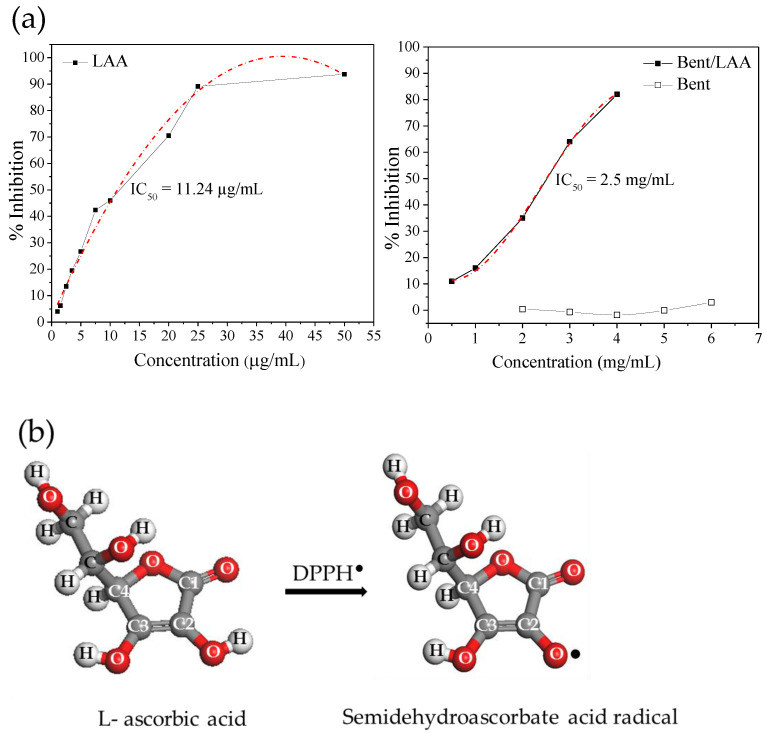
Antioxidant capacity for LAA, Bent and the composite Bent/LAA, including the IC_50_ values for each sample (**a**), and a sketch of the reaction between LAA and DPPH^●^ (**b**).

**Table 1 pharmaceutics-15-01171-t001:** Chemical composition in oxides percentage of purified Bent raw material.

Sample	SiO_2_	Al_2_O_3_	Fe_2_O_3_	Na_2_O	MgO	CaO	TiO_2_	K_2_O	MnO	LOI ^1^
Bent	48.86	19.05	9.42	2.81	2.26	1.71	0.82	0.21	0.07	12.68

^1^ LOI—loss on ignition.

## Data Availability

The data presented in this study are available on request from the corresponding author.

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
