# Peer review of "Antioxidant Efficacy and “In Vivo” Safety of a Bentonite/Vitamin C Hybrid"

_pharmaceutics, 2023, doi:10.3390/pharmaceutics15041171_

Round 1

Reviewer 1 Report

This is an interesting manuscript that is submitted by Hernández et al. in Pharmaceutics.

It reports on the insertion of ascorbic acid molecules into the interlayers of bentonite clays, more specifically, into its main crystalline phase, which is the swelling clay mineral, montmorillonite.

Although ascorbic acid intercalation and immobilization by bentonite has already been reported in the literature, the new thing is that its dermal toxicity, antioxidant efficiency and photostability has not yet been investigated in details. In this respect, I think that the study is valuable and should be published in Pharmaceutics, after some revision, especially focusing on the interpretation of some data.

Technical remarks:

-    -     line 254: “Quart” must be “quartz”

-  -      line 277: “interactions adsorbent/adsorbate interactions that” remove the first “interactions”

-  -       line 345: another stage of dihydroxylation

Remarks on science:

-  -       “LAA incorporation on montmorillonite was reported”. Although the authors provide citations regarding this statement, it would be also very interesting to compare the present results to the already published ones, in terms of a few aspects: which methods have been use previously, and what was the loading efficiency, etc. Of course, this is not a literature review, but there are these three papers which are key references. I suggest to work on comparison and include it into the manuscript

-  -       “the supernatant was eliminated by decantation.”. Actually, it is unclear, why this was done. The montmorillonite part is very likely easily dispersible, especially when it is in its Na-form, when it exfoliated to single sheets that may not settle down. Why were they discarded? Also, why was the other mineral particles (quartz) not removed by gravitational separation from the solid to be used for the formulation?

-  -       “LAA aqueous solution at 9 mg/mL was put in contact with 100 mg of Bent in powder and continuous stirring during 24 h at room temperature”: here, my problem is, whether the LAA was decomposed partially during the long time of 24 hours in the solution. The authors also noted that LAA is only stable in fairly acidic solution, so here, it is very likely that part of the LAA was not removed from the liquid phase by adsorption and intercalation, but by the chemical decomposition. This is also likely because the adsorbed amount of 450 mg ± 50 of LAA per gram of Bent looks unrealistically high. How can be the adsorbed amount be so high, if the LAA adsorbs in a monolayer form between layers? Also, there are other mineral phases that may not adsorb any LAA (quartz). In fact, some realistic picture on the incorporation of such a high level of LAA should be provided.

-   -      “several alternative scenarios regarding the LAA adsorption on the Bent can be suggested:” I checked these, but I am unsure about some of them. (1) “its smaller dimension is perpendicular to the clay layers”: what does it mean “smaller” here? There are 3 different dimensions, is that the smallest one? (2) “the preferential orientation of some functional groups”: this is unclear to me and likely to many readers. A scheme may help to imagine what do the authors mean. (3) “it could be the LAA molecules adsorption” possibly yes, but what whould be the driving force for adsorption? (4) “effect of two water layers intercalated” is not very likely, because the van der waals diameter of water molecules are around 26-28 Angstroms, and this would make up almost 6 angstroms of increase of interlayer distance. So, a bilayer of water molecules is not very likely.

-    -     “It may indicate the Bent photoprotective effect onto LAA structure resulting from the small drug degradation occurring when the composite was exposed to UV irradiation.”. This is indeed possible and very interesting question. My problem is: how can we be sure if the clay particles indeed protected the particles from photodegradation. What the experiment shows is that, by some rate, the LAA is gradually decomposing upon illumination. But, is that also happening by the same rate with LAA in its pure crystalline form? I know it may be challenging to measure this, but without some comparative experiments for the inserted and “original”, crystalline forms of LAA, the authors should not make a firm conclusion on this point.

-  -       “contribution of electrostatic interactions in the composite material formation.”: How would this be possible? Clay layers are negatively charged, and also ascorbic acid. Only the edge sites, depending on pH, can be positively charged. I think, there is another possible explanation why the zeta potential values of the composite are less negative than for the pristine clay: ionic strength effect. The authors were careful to consider pH changes of the surface charge, but zeta potential is also sensitive to the ionic composition of the medium: Even if the surface charge is the same, is the salt concentration increases in the solution, this will screen more efficiently the surface charge and provides a loss of the zeta potential. So the conclusion about electrostatic interaction looks fairly speculative with the present experimental data.

Author Response

Answer to Reviewer #1 report comments:

Technical remarks:

- -     line 254: “Quart” must be “quartz”

-  -    line 277: “interactions adsorbent/adsorbate interactions that” remove the first “interactions”

-  -    line 345: another stage of dihydroxylation

These remarks were corrected in the revised version of the manuscript.

Remarks on science:

-  -       “LAA incorporation on montmorillonite was reported”. Although the authors provide citations regarding this statement, it would be also very interesting to compare the present results to the already published ones, in terms of a few aspects: which methods have been use previously, and what was the loading efficiency, etc. Of course, this is not a literature review, but there are these three papers which are key references. I suggest to work on comparison and include it into the manuscript.

ANSWER: We thank the referee for pointing out this matter. Without a doubt, a comparison of results would have been very interesting. However, it is not a simple matter to include within a limited space. In two of the references cited near the end of the introduction of the manuscript [1,2], the authors basically discuss the use of natural montmorillonite (Mt) and acid treated Mt in the intercalation and stabilization of ascorbic acid (LAA) for potential uses (not defined). In the first study, despite that a new method is declared in the last part of the introduction, is not properly described later in the article. The data corresponding to LAA load on the clay is not reported either. The second article focuses on the stabilization of the activity of LAA in dissolution at different natural and modified Mt concentrations. The third reference cited in our manuscript [3], studies the employment of natural and cation modified smectites  as supports for pollutants removal, where the LAA was chosen as model emergent contaminant. In this work the type of smectite is not identified. Although in this investigation different parameters ―pH, LAA initial concentration from and time― were optimized, a comparison with our results does not seem suitable due to the following elements: 1) No details are given on the type material evaluated, 2) They involve very low LAA initial concentrations (from 0 to 0.1 mg/mL), and 3) At the optimal values of pH = 5 and pH = 8, the LAA species (pka = 4.2) are negatively charged.  In principle, they must not interact with a negatively charged surface as would be the case with our clay (Bent) (now Fig. 6 of our manuscript). Additionally, at basic pH the LAA degradation is favored damaging its stability and antioxidant capacity.

For these reasons, the comparison with these papers was not included into the manuscript. However, and based on this suggestion, the authors has decided to include part of these comments in the Supplementary materials (S1).

-  -  “the supernatant was eliminated by decantation.”. Actually, it is unclear, why this was done. The montmorillonite part is very likely easily dispersible, especially when it is in its Na-form, when it exfoliated to single sheets that may not settle down. Why were they discarded? Also, why was the other mineral particles (quartz) not removed by gravitational separation from the solid to be used for the formulation?

ANSWER: It is true that the Mt has a high swelling capacity, which leads to think that the exfoliation/delamination processes would be favored in aqueous dispersions. It is known that the smectite−water systems are colloidal suspensions formed by single platelets or small aggregates. Note also that the hydration regime of montmorillonite (Mt) in the presence of water contents depends on the nature of the exchange cation: in homoionic calcic (Ca-Mt) samples the swelling is limited to crystalline swelling, while the homoionic Na-Mt exhibits two regimes, crystalline swelling and osmotic swelling [4,5]. However, in the literature it is well established hat certain smectites are a rare example of such spontaneous delamination through progressive osmotic swelling [5]. Furthermore, in multicatioic natural samples (like our case) submitted to purification processes during short cycles of dispersion of the solid in water and high solid-liquid ratios, the decantation of a first supernatant with a very low clay proportion and sedimentation of the clay itself is favored. Regarding this, some related comments were included in the new version of the manuscript (2.2.1. section) in order to clarify the ideas.

Regarding the removing of quartz by gravitational separation, this procedure is not always suitable. For example, in the case of mineral phases with similar densities, it is ineffective [6]. In our case, it was not used to remove quartz (Q) from montmorillonite (Mt) since the specific weights of both are very close to each other (2.62 g/cm3 and 2.35 g/cm3 as average for the Q and Mt, respectively) [7]. In addition, the XRD results allowed verifying the presence of Mt main phase in the sample, as well as a very low proportion of Q regarding to clay mineral. Moreover, its potential use in formulations was confirmed by the toxicological studies performed on the Bent sample.

   - -   “ “LAA aqueous solution at 9 mg/mL was put in contact with 100 mg of Bent in powder and continuous stirring during 24 h at room temperature”: here, my problem is, whether the LAA was decomposed partially during the long time of 24 hours in the solution. The authors also noted that LAA is only stable in fairly acidic solution, so here, it is very likely that part of the LAA was not removed from the liquid phase by adsorption and intercalation, but by the chemical decomposition. This is also likely because the adsorbed amount of 450 mg ± 50 of LAA per gram of Bent looks unrealistically high. How can be the adsorbed amount be so high, if the LAA adsorbs in a monolayer form between layers? Also, there are other mineral phases that may not adsorb any LAA (quartz). In fact, some realistic picture on the incorporation of such a high level of LAA should be provided.

ANSWER: We agree that special attention should be given to LAA stability in aqueous solution. It depends upon many factors such as oxygen, temperature, light, storage conditions and pH [8,9]. As the referee mentions, 100 mg of Bent were put in contact with a LAA aqueous solution at 9 mg/mL, and continuous stirring during 24 h at room temperature in our case. The pH value of the clay/drug suspension spontaneously reached was around 3.5. It is known that the largest LAA molecules stability in aqueous medium takes place at acid pH below its pKa1 (4.2), as mentioned in the manuscript, and where the LAA is mainly in a non-ionized form (H2LAA). As the pH increases, the concentrations of anionic species rises (HLAA- at pH 7.4 and LAA2- at pH > pKa2 = 11.6) [10]. From the pH values above its pKa2, redox processes of LAA species will be favored affecting the drug stability, as well as its antioxidant capacity [2]. Furthermore, in order to protect the Bent/LAA suspensions from the effect of light, amber containers with aluminum paper as extra protection were used. At the end of the process and after centrifugation, a total transparency of the supernatant was also observed. It confirms the non-degradation of the drug since no color changes were evidenced. Considering these facts, the proved ability of Mt to stabilize the LAA in dissolution [2], and the shown antioxidant activity of our Bent/LAA hybrid, it is possible to ratify the LAA load by adsorption /intercalation on the Bent clay and not by its chemical decomposition.

Considering the referee’ comments, new information relative to the interaction process in acid medium to preserve the LAA stability in aqueous dispersion was included in the revised version of the manuscript (2.2.3. and 3.2. sections, first and second paragraphs, respectively).

Regarding the adsorbed amount of LAA per gram of Bent, we have discussed it in more detail in section 3.2 (third, fourth and fifth paragraphs), and a new figure was included (now Figure 3). If we consider the dimensions of the unit cell of Mt (a = 0.52, b = 0.90, c = 1.00 nm [7]), those of the LAA molecule (0.74 in length, 0.50 in wide and 0.34 nm in thickness following the more stable conformers [11]), the existence of free rotation carbons in the drug structure, as well as the variation of the d-value in the XRD pattern of the composite, we conclude that one LAA molecule per unit cell of Mt can easily accommodate in the interlayer space. On the other hand, if we consider that 1 g of Bent contains 7.69 x 1020 unit cells and that the LAA load measured by UV spectroscopy is about 450 mg per gram of Bent, in one Bent unit cell there are 5.85 x 10-19 mg of LAA. Then, if one LAA particle weights 2.93 x 10-19 mg, we would have around of 2 LAA molecules per unit cell. So, we hypothesize that at least one of these two LAA molecules is located in the interlayer space, and the other is adsorbed on the clay surface and into the material mesoporosity/macroporosity. This scenario semi-quantitatively explains the relatively large load of LAA on the Bent.

- -   “several alternative scenarios regarding the LAA adsorption on the Bent can be suggested:” I checked these, but I am unsure about some of them. (1) “its smaller dimension is perpendicular to the clay layers”: what does it mean “smaller” here? There are 3 different dimensions, is that the smallest one? (2) “the preferential orientation of some functional groups”: this is unclear to me and likely to many readers. A scheme may help to imagine what do the authors mean. (3) “it could be the LAA molecules adsorption” possibly yes, but what whould be the driving force for adsorption? (4) “effect of two water layers intercalated” is not very likely, because the van der waals diameter of water molecules are around 26-28 Angstroms, and this would make up almost 6 angstroms of increase of interlayer distance. So, a bilayer of water molecules is not very likely.

ANSWER: The referee’s suggestions were taken into account. The authors agree that a scheme is useful to illustrate the various alternative scenarios regarding the LAA adsorption on the Bent. A more detailed description of the section, a new reference and a new figure (now Figure 3) were introduced in the revised version of the manuscript (third, four and fifth paragraphs of the 3.2 section).                                                   

UV spectroscopy and X-ray diffraction (XRD) results allowed corroborated the LAA incorporation on the Bent clay, evidencing an additional increment of interlayer distance for the Bent/LAA composite. However, the complexity of the system encouraged us to consider a number of possibilities that may explain the clay/drug interactions: the more evident, no doubts, is the LAA intercalation in the interlayer space but other host-guest interactions should not be ignored, which would also explain the LAA load on the Bent. Regarding the Van der Waals diameter of water molecules, we assume that in the values given by the referee a typographic mistake was committed. The increment of the interlayer space by the effect of one water layer intercalated typically is about 0.28 nm [4,12]. Thus, when two water layer are present, an increment of the interlayer space about 0.56 nm is to be expected. It will support the scenario where presence of two water molecules and the presentation of the drug at the entrance of the interlayer take place.

-    -     “It may indicate the Bent photoprotective effect onto LAA structure resulting from the small drug degradation occurring when the composite was exposed to UV irradiation.”. This is indeed possible and very interesting question. My problem is: how can we be sure if the clay particles indeed protected the particles from photodegradation. What the experiment shows is that, by some rate, the LAA is gradually decomposing upon illumination. But, is that also happening by the same rate with LAA in its pure crystalline form? I know it may be challenging to measure this, but without some comparative experiments for the inserted and “original”, crystalline forms of LAA, the authors should not make a firm conclusion on this point.

ANSWER: Thank you for finding very interesting our suggestion. The referee is right: a direct measurement of the Bent photoprotection onto LAA structure is not given, since there is not a procedure for such studies in the solid state, as far as we know. However, from the small shift observed in the Bent 001 basal reflection for the irradiated samples it is possible to infer a slight degradation of the intercalated LAA. It should be noted that this is really not representative of the decay rate of the LAA, especially when dealing with intercalated species. The effect of Bent on the LAA decay rate has been studied in the literature for stability enhancement of LAA [2]. In that case, the LAA concentration in solution for several Bent amounts as a as function of time is evaluated. The results indicated that when the Bent is in the dissolution medium, the LAA decomposition in dissolution noticeably decreases. It is well know that the LAA degradation occurs basically when it is in dissolution [8]. Due to that, a comparison with the LAA in pure crystalline form (or with Bent/LAA hybrid samples, where the LAA species do not have a crystalline ordering in the composite) does not seem to be appropriated.         

A related discussion was included in the revised manuscript (sixth paragraph, section 3.2).

-  -       “contribution of electrostatic interactions in the composite material formation.”: How would this be possible? Clay layers are negatively charged, and also ascorbic acid. Only the edge sites, depending on pH, can be positively charged. I think, there is another possible explanation why the zeta potential values of the composite are less negative than for the pristine clay: ionic strength effect. The authors were careful to consider pH changes of the surface charge, but zeta potential is also sensitive to the ionic composition of the medium: Even if the surface charge is the same, is the salt concentration increases in the solution, this will screen more efficiently the surface charge and provides a loss of the zeta potential. So the conclusion about electrostatic interaction looks fairly speculative with the present experimental data.

ANSWER: The authors agree with the referee concerning the ionic strength effect on Mt zeta potential measurements. It should be note that the zeta potential of Mt particles does not change significantly with the pH [13]. Zeta potential measurements for both samples (Bent and Bent/LAA) were performed under the same conditions: H2O purified with a conductivity around 0.056 µS/cm was employed as dispersion medium, without the extra addition of salts to the dissolution. Hence, the presence of the drug is the only one difference between them. In particular, at pH = 3.5 where the clay/drug interaction takes place, the LAA molecule is mainly in a non-ionized form (H2LAA, pKa1 = 4.2), i.e., in its neutral molecular form by absence of charge. In non-symmetric molecules with polar groups, like the case of the LAA, electronic densities are located on the more electronegative elements generating dipoles which can interact with the Mt negative charge. To refer to these interactions or Van der Waals forces, we simply use the term “electrostatic interactions” in the manuscript.    

These facts and a new reference were incorporated in the new version of the manuscript (sixth paragraph of the section 2.2.4. and last paragraph of the section 3.2.).

References

  1. Chen, B.-Y.; Lee, Y.-H.; Lin, W.-C.; Lin, F.-H.; Lin, K.-F. Understanding the characteristics of L-ascorbic acid-montmorillonite nanocomposite: Chemical structure and biotoxicity. Biomed. Eng-App. Bas. C. 2006, 18.
  2. Lee, Y.-H.; Chen , B.-Y.; Lin, K.-Y.; Lin, K.-F.; Lin, F.-H. Feasibility study of using montmorillonite for stability enhancement of L-ascorbic acid. J. Chin. Inst. Chem. Eng. 2008, 39, 219-226.
  3. Anouar, F.; Elmchaouri, A.; Taoufik, N.l.; Rakhila, Y. Investigation of the ion exchange effect on surface properties and porous structure of clay: Application of ascorbic acid adsorption. J. Environ. Chem. Eng. 2019, 7, 103404.
  4. Sun, L.; Tanskanen, J.T.; Hirvi, J.T.; Kasa, S.; Schatz, T.; Pakkanen, T.A. Molecular dynamics study of montmorillonite crystalline swelling: Roles of interlayer cation species and water content. Chem. Phys. 2015, 455, 23-31.
  5. Stöter, M.; Rosenfeldt, S.; Breu, J. Tunable Exfoliation of Synthetic Clays. Annual Review of Materials Research 2015, 45, 129-151.
  6. Ciftci, H. An Introduction to Montmorillonite Purification. In Montmorillonite Clay, Faheem, U., Ed.; IntechOpen: Rijeka, 2021; p. Ch. 5.
  7. Mineralogy-Database. Available online: http://webmineral.com/data/Montmorillonite.shtml (accessed on March 2023).
  8. Caritá, A.C.; Fonseca-Santos, B.; Shultz, J.D.; Michniak-Kohn, B.; Chorilli, M.; Leonardi, G.R. Vitamin C: One compound, several uses. Advances for delivery, efficiency and stability. Nanomed.: Nanotechnol. Biol. Med. 2020, 24, 102117.
  9. Ravetti, S.; Clemente, C.; Brignone, S.; Hergert, L.; Allemandi, D.; Palma, S. Ascorbic Acid in Skin Health. Cosmetics 2019, 6, 1-8.
  10. Pilarski, B.; Wyrzykowski, D.; MÅ‚odzianowski, J. A new approach for studying the stability and degradation products of ascorbic acid in solutions; Preptint in Research Square 2022.
  11. Milanesio, M.; Bianchi, R.; Ugliengo, P.; Roetti, C.; Viterbo, D. VitaminC at 120K: experimental and theoretical study of the charge density. J. Mol. Struc-Theochem. 1997, 419, 139-154.
  12. Diamond, S.; Kinter, E.B. Characterization of montmorillonite saturated with short-chain amine cations: 1. Interpretation of basal spacing measurements. Clays and Clay Miner. 1961, 10, 163-173.
  13. Zaka, E.E.; Güler, C. The effects of electrolyte concentration, ion species and ph on the zeta potential and electrokinetic charge density of montmorillonite. Clay Miner. 2006, 41, 853-861.

Reviewer 2 Report

The manuscript Pharmaceutics-2212031 by D. Hernández et al. describes preparation of bentonite (Bent) clay incorporated with L-ascorbic acid (LAA), characterization of this composite material using UV and FTIR spectroscopy, XRD, TG/DTG and zeta potential measurements, ophthalmic irritability and acute dermal toxicity assays of pure bentonite, as well as photostability and antioxidant capacity tests of prepared composite.

The study adds some new knowledge to this field. I expect the work will be of interest to the scientific community, prepared bentonite/LAA composite showed to be promising for potential application in skin care formulations. The abstract is informative and the reference are relevant to the topic. The experiments are precisely described. The results are well presented. The subject is suitable for the journal Pharmaceutics. I do recommend the manuscript for publishing in Pharmaceutics, but after revision.

I have the following comments on the manuscript:

The determined LAA load onto/into bentonite seems to be quite high: 450 ± 50 mg per 1 g of bentonite (that is 0.5 g LAA per 1 g bentonite). If the incorporation of LAA is that high, I would expect that LAA effect would be more visible on XRD patterns and FTIR spectra. Also, if this is the load of LAA than it is not in line with the concluded protective effect of bentonite on LAA antioxidant activity. That is, the conclusion related to antioxidative activity and protective effect of bentonite should be clarified a little better; that is, better describe Figure 6. If the load is 0.5 g LAA per 1 g bent, then the bent/PLLA composite has significantly higher antioxidant activity than pure bent (which has no activity), but still significantly lower compared to pure LAA (for pure LAA IC50 is 11,2 mm/ml, while for bent/LAA is 2,5 mg/ml that is around 1 mg/ml LAA if the load is as stated).

The conclusions about photoprotective effect of bentonite onto LAA is very indirect. Put XRD results of UV irradiated pure LAA for comparison (it can be included into the Supplementary material and just commented in the text).

In addition, some evidence of the in vivo toxicity studies of pure bentonite clay can be given in the Supplementary material.

In Figure 6 (left), the value of the maximum temperature should be corrected, 518 oC should be written instead of 228 oC.

Correct line 334 page 8 “lesser wavenumber” to “lower wavenumber”.

Correct line 368 page 9 “smaller temperature” to “lower temperature”.

I recommend the manuscript for publication in Pharmaceutics after the authors have revised the manuscript in accordance with the above comments.

Author Response

Answer to Reviewer #2 report comments:

- The determined LAA load onto/into bentonite seems to be quite high: 450 ± 50 mg per 1 g of bentonite (that is 0.5 g LAA per 1 g bentonite). If the incorporation of LAA is that high, I would expect that LAA effect would be more visible on XRD patterns and FTIR spectra.

ANSWER: Indeed, the LAA load is about 0.45 g per 1 g of Bent. As is shown in the XRD pattern for the Bent/LAA hybrid (Figure 2 A), the LAA intercalation in the interlayer space, provoked a shift visible (around Δd = 0.63 nm) of the 001 basal reflection plane of Mt. The high LAA load can be explained as follows: 1 g of Bent contains 7.69 x 1020 unit cells and LAA load measured by UV spectroscopy is about 450 mg per gram of Bent, and in one Bent unit cell there is 5.85 x 10-19 mg of LAA.  Then, if one LAA particle weight 2.93 x 10-19 mg, we would have around of 2 molecules per unit cell of Bent. So, we hypothesize that at least one of these two LAA molecules is located in the interlayer space, and the other is adsorbed on the clay surface and into the material mesoporosity/macroporosity [1]. In such case, a new peak in the XRD pattern will be not observed, since the drug is not in its crystalline form on the material.  In addition, these results were corroborated with those obtained by ATR, where modifications in characteristics vibration modes of the LAA were evidenced, in particular for the band at 1631 cm-1, as discussed in the manuscript.

The XRD relative changes have been better discussed, with the inclusion of a new Figure (now Figure 3), in the new version of the manuscript (third, four and fifth paragraphs, section 3.2), following the suggestions of both referees.

- Also, if this is the load of LAA than it is not in line with the concluded protective effect of bentonite on LAA antioxidant activity. That is, the conclusion related to antioxidative activity and protective effect of bentonite should be clarified a little better; that is, better describe Figure 6.

- The conclusions about photoprotective effect of bentonite onto LAA is very indirect. Put XRD results of UV irradiated pure LAA for comparison (it can be included into the Supplementary material and just commented in the text).

ANSWER: Due to the relation between these issues, we are addressing them together.

It is true that the antioxidative activity and the protective effect of bentonite should be clarified. In this sense, the small shift observed in XRD (001 basal reflection of the MT phase) for the irradiated samples, constitutes an indicative of the slight degradation of the intercalated LAA. It suggests the Bent photoprotective action onto LAA structure when the Bent/LAA hybrid was exposed to UV radiation.

Certainly, the photoprotective effect of Bent onto LAA was evaluated in indirect way. However, it should be noted that there is not a procedure for such studies in solid state, as far as we know, and that the LAA degradation occurs basically when the drug is in dissolution [2]. Therefore, a comparison between the LAA in crystalline form and those Bent/LAA composites ― where the LAA species do not have a crystalline ordering in the composite―, does not seem to be appropriated.

Based on the referee’s recommendations, the related discussion has been changed in the revised manuscript (sixth paragraph, section 3.2).

- If the load is 0.5 g LAA per 1 g bent, then the bent/PLLA composite has significantly higher antioxidant activity than pure bent (which has no activity), but still significantly lower compared to pure LAA (for pure LAA IC50 is 11,2 mm/ml, while for bent/LAA is 2,5 mg/ml that is around 1 mg/ml LAA if the load is as stated).

ANSWER: We agree with the referee, indeed the antioxidant activity is lower for Bent/LAA hybrid than for pure LAA. In such case, we would like to underline that LAA preserves its antioxidant activity even after the intercalation into Bent clay. However, the apparent worsening of the activity it could be explained by taking into account the possible interactions Bent/LAA: as it is known, the antioxidant activity of LAA is mainly due to the action of the –hydroxyl group (–OH) linked to C2 (see Figure 7b of the manuscript). After interaction with Bent, this -OH could establish hydrogen bonds with clay and thus it is not available anymore to exert the antioxidant activity. The existence of hydrogen bonding interaction between organic molecules and clays was indeed reported in literature [3,4].Therefore, it is possible to conclude that Bent perform some protective effect on LAA antioxidant activity.

In order to clarify this point, these elements were introduced in the revised manuscript (section 3.3.), with the inclusion of a schema in Figure 7 for a better visualization of the process.

- In addition, some evidence of the in vivo toxicity studies of pure bentonite clay can be given in the Supplementary material.

Flowing the referee`s suggestion, it has been given in the Supplementary material.

- In Figure 6 (left), the value of the maximum temperature should be corrected, 518 oC should be written instead of 228 oC.

The error was corrected in the new version of the manuscript.

- Correct line 334 page 8 “lesser wavenumber” to “lower wavenumber”.

- Correct line 368 page 9 “smaller temperature” to “lower temperature”.

Both remarks were corrected in the revised version of the manuscript.

References

  1. Fabrice, S.; D., J.-M.; Renaud, D.; Olivier, B.; Michel, J.; Isabelle, B.; Henri, V.D. Hydration sequence of swelling clays: Evolutions of specific surface area and hydration energy. Journal of Colloid and Interface Science 2009, 333, 510-522.
  2. Caritá, A.C.; Fonseca-Santos, B.; Shultz, J.D.; Michniak-Kohn, B.; Chorilli, M.; Leonardi, G.R. Vitamin C: One compound, several uses. Advances for delivery, efficiency and stability. Nanomed.: Nanotechnol. Biol. Med. 2020, 24, 102117.
  3. White, J.L.; Hem, S.L. Pharmaceutical aspects of clay-organic interactions. Ind. Eng. Chem. Prod. Res. Dev 1983, 22, 665-671.
  4. Yang, J.-H.; Lee, J.-H.; Ryu, H.-J.; Elzatahry, A.A.; Alothman, Z.A.; Choy, J.-H. Drug–clay nanohybrids as sustained delivery systems. Appl. Clay Sci. 2016, 130, 20-32.

Round 2

Reviewer 1 Report

The authors made a good argument and a fair improvement of this manuscript. I think it is good quality piece of work and can now be published.

Reviewer 2 Report

The revised manuscript Pharmaceutics-2212031 is significantly improved. The authors introduced all the necessary changes and revised the manuscript according to the comments. I do recommend publishing the paper in the present form.